# CD46 Is a Protein Receptor for Human Adenovirus Type 64

**DOI:** 10.3390/v16121827

**Published:** 2024-11-25

**Authors:** Eugene Y. Wu, Alexander M. Robertson, Hanglin (Henry) Zhu, Corina Stasiak, Laura A. Murray-Nerger, Emily Romanoff, Jesse Woon, Beth A. Bromme, Jason G. Smith

**Affiliations:** 1Department of Biology, University of Richmond, Richmond, VA 23173, USA; alex.robertson@richmond.edu (A.M.R.); henry.zhu@richmond.edu (H.Z.); eromanoff@lifespan.org (E.R.);; 2Department of Microbiology, University of Washington School of Medicine, Seattle, WA 98109, USAjgsmith2@uw.edu (J.G.S.)

**Keywords:** adenovirus, epidemic keratoconjunctivitis, CD46, membrane cofactor protein, receptor

## Abstract

Certain species D human adenoviruses (HAdV-D19, -D37, and -D64) are causative agents of epidemic keratoconjunctivitis. HAdV-D37 has previously been shown to bind CD46 (membrane cofactor protein) and sialic acid as adhesion receptors. HAdV-D64 is genetically highly similar to HAdV-D37, with an identical fiber protein sequence, but differs substantially in its penton base and hexon proteins, two other major capsid components, due to genetic recombination. Here, we demonstrate that, like HAdV-D37, HAdV-D64 virions bind directly to CD46 and that CD46 and sialic acid also function as receptors for HAdV-D64 on multiple cell types. Expression of CD46 on CD46-negative cells conferred susceptibility to HAdV-D64 entry. Specifically blocking HAdV-D64 binding to CD46 on the host cell surface strongly inhibits viral entry and gene delivery into multiple cell lines that represent target tissues. We show that CD46 is expressed on human conjunctival epithelial cells and directly binds to the HAdV-D64 virion. Our results suggest that HAdV-D64 may be used to deliver genes to target conjunctival cells and that interrupting HAdV-D64 entry through its interaction with CD46 may prevent or lessen adenovirus-associated ocular disease.

## 1. Introduction

Human adenoviruses (HAdVs) are non-enveloped, double-stranded DNA viruses associated with a variety of human infections. The external surface of the adenovirus capsid is mainly comprised of three proteins: (1) the hexon, the main constituent of the T = 25 icosahedral shell; (2) the penton base, a pentameric complex at each icosahedral vertex; and (3) the fiber, a trimeric protrusion from the penton vertices. For most HAdVs, the distal knob domain of the fiber binds to one of several known primary receptors to adhere to the target cell surface, and penton base proteins bind cell surface integrins to induce endocytosis for entry (for reviews, see [1,2]). 

The over 100 adenovirus types have been classified into seven species (A-G), with approximately half of the known types in species D [3]. Genetically homologous species D types 8, 19, 37, 53, 54, 56, and 64 are associated with epidemic keratoconjunctivitis (EKC), a severe ocular infection characterized by conjunctival swelling, excessive watering of the eye, and sometimes corneal infiltrates [3]. EKC clinical isolates previously identified as HAdV-D19 were reclassified as HAdV-D64 after genomic sequencing and sequence comparisons showed high homology to the prototype HAdV-D19 (HAdV-D19p), isolated from a trachoma patient in Saudi Arabia [4], only in their hexon genes [5]. In addition, HAdVs-D19, D37, D56, and D64 have occasionally been associated with genitourinary infections such as cervicitis and urethritis [6,7,8,9,10]. Beyond causing human disease, genetically modified adenoviruses are important vectors for gene delivery, gene therapy, and vaccine design (for reviews, see [11,12,13]). 

One of the primary determinants of vector targeting and viral tropism is the presence of entry receptors on the surface of target cells. Previous studies have identified CD46 (also known as membrane cofactor protein) as an adhesion receptor for HAdV-D37 [14,15], and sialic acid as an adhesion receptor for HAdV-D37 and D19p [16,17,18,19]. A sialic-acid-containing glycan from the Gd1a ganglioside has been identified as a receptor for HAdV-D37 on human corneal epithelial cells [20]. CD46 has also been identified as an adhesion receptor for several species B adenoviruses (types 3, 11, 14, 16, 21, 34, 35, 50) [21,22,23,24], some of which cause ocular infections [3]. 

Because sialic acid and CD46 are important receptors for multiple adenovirus types and species that infect the eye, probing the interaction between these receptors and the adenovirus capsid is important for understanding viral entry and tropism. While adenovirus fiber proteins bind directly to some receptors such as the coxsackievirus-adenovirus receptor [25] and sialic acid [19,20], several species D adenoviruses were reported to bind CD46 using their hexon proteins, including EKC-associated HAdV-D56 but not HAdV-D37 [26]. The genome of HAdV-D37 is 98.6% identical to that of HAdV-64, but these two EKC-associated HAdVs differ substantially in their hexon and penton base genes [5], creating uncertainty about whether HAdV64 also uses CD46 to enter target cells and, if so, through which capsid protein.

HAdVs D37 and D64 have identical fiber protein sequences and are associated with the same human diseases. Thus, we hypothesized that HAdV-D37 and -D64 use the same receptors on the surface of target genes for adhesion. Here, we use an HAdV-D64 virus in which the non-essential viral E3 gene has been substituted by a green fluorescent protein reporter gene to test whether HAdV-D64 uses sialic acid and/or CD46 on the surface of several cell types. We show that blockage or removal of sialic acid and/or CD46 inhibits HAdV-D64 entry, that expression of CD46 on non-susceptible cells confers susceptibility to HAdV-D64 entry, and that HAdV-D64 virions bind directly to CD46, establishing that HAdV-D64 uses CD46 as a receptor. 

## 2. Materials and Methods

### 2.1. Culture of Cell Lines and Adenoviruses

Adherent human conjunctival epithelial (HCjE) cells [27,28] were obtained from Dr. James Chodosh (Massachusetts Eye and Ear and Harvard Medical School; currently at University of New Mexico School of Medicine) and grown in supplemented Keratinocyte Serum-Free Medium (KSFM; ThermoFisher Scientific, Waltham, MA, USA). Adherent A549 lung carcinoma cells (ATCC, Manassas, VA, USA), HeLa cervical carcinoma cells (ATCC), and Chinese hamster ovary (CHO) cells stably transfected with CD46 expression plasmids [29] were cultured in Advanced Dulbecco’s Modified Eagle’s Medium with 3% fetal bovine serum or Dulbecco’s Modified Eagle’s Medium with 10% fetal bovine serum. 

Replication-competent HAdV-D64 virus containing the enhanced green fluorescent protein gene driven by a cytomegalovirus promoter in place of E3 (Ad64.eGFP) was propagated as previously described [30] and purified by cesium chloride density ultracentrifugation. Viral concentration was estimated using UV spectrophotometry, and purity was confirmed using sodium dodecyl sulfate polyacrylamide gel electrophoresis (Bio-Rad, Hercules, CA, USA).

### 2.2. Recombinant Protein Expression

The gene encoding isoform C of the soluble extracellular domain of CD46 (sCD46-C) was cloned into pTT5. 293-6E cells were transfected with the resulting plasmid and then grown in Freestyle 293 Expression Medium. sCD46-C was purified using HisTrap FF chromatography from cell culture medium (molecular cloning, overexpression, purification, and gel electrophoresis performed by Genscript Biotech (Piscataway, NJ, USA)).

### 2.3. Adenovirus-Mediated Gene Delivery Assay for Viral Entry

CHO cells containing CD46 expression plasmids were grown in 24-well plates at a density of 50,000 cells per well. The cells were washed with Tris-buffered saline (20 mM Tris-HCl, pH 7.5, 120 mM NaCl) and then infected with 5 × 10^8^ particles of Ad64.eGFP in Tris-buffered saline (TBS) for 1 h at RT. The cells were washed in TBS and cultured overnight at 37 °C. Gene delivery was assessed by flow cytometry after detachment with Trypsin/EDTA using a BD Accuri C6 flow cytometer (Becton Dickinson, Franklin Lakes, NJ, USA). The cells were counted as infected if their fluorescence values in the FITC-A (green) channel were higher than the highest values for uninfected cells. HeLa cells were grown in 24-well plates at a density of 25,000 cells per well. The cells were washed with TBS prior to incubation with 2.5 × 10^8^ Ad64.eGFP diluted in TBS alone, TBS plus 20 µg/mL N-19 CD46 antibody (Santa Cruz Biotechnology, Santa Cruz, CA, USA), TBS plus 1 mM calcium chloride, or TBS plus 1 mM calcium chloride and 20 µg/ml N-19 CD46 antibody for one hour at room temperature. Virus was removed, and the cells were cultured overnight at 37 °C prior to detachment and flow cytometry analysis. To test the influence of neuraminidase on gene delivery, 50,000 HeLa cells per well were pretreated with 10 mU neuraminidase from *Vibrio cholerae* (Sigma-Aldrich, St. Louis, MO, USA) for 1 h in TBS prior to infection with 4 × 10^7^ Ad64.eGFP particles for one hour at room temperature. To test the influence of ethylene glycol tetraacetic acid (EGTA, Sigma-Aldrich), 50,000 HeLa cells were infected with 4 × 10^7^ Ad64.eGFP particles in TBS plus 10 mM EGTA for one hour at room temperature. 

HCjE cells were grown overnight at 37 °C in a 24-well plate containing 10,000 cells per well and then infected with 2 × 10^7^ particles of Ad64.eGFP overnight at 37 °C in 250 µL of KSFM with or without 3 mM EGTA, which is 25 times the concentration of divalent calcium in KSFM. The cells were detached using 250 µL of TrypLE Express (ThermoFisher Scientific) for 10 min at 37 °C and transferred to wells in a 96-well plate to increase density. The cells were imaged in 4 non-overlapping locations using a ZOE Fluorescent Cell Imager (Bio-Rad) using brightfield and green fluorescence channels. Infected (cells fluorescing with higher intensity than uninfected cells) and total cells were counted in the 0.7 mm^2^ viewfields (viewfields with >50 cells were counted using ImageJ software (https://imagej.net/ij/; accessed on 23 October 2024)). For CD46 blocking, sCD46-C was added to virus-laden SFM at a final concentration of 0.35 µM and incubated at room temperature for 5 min before being added to the wells. For sialic acid cleavage, neuraminidase (Sigma-Aldrich) treatment of HCjE cells prior to viral infection (7.2 × 10^7^ particles per well on 50,000 cells per well) involved a final concentration of 31 milliunits incubated for 1 h at 37 °C. The cells the following day were detached using TrypLE Express at 37 °C and analyzed by flow cytometry. 

Fiber knob competitive inhibition of adenovirus entry on A549 cells was performed in 96-well plates. The wells were pre-treated with HAdV-B16 fiber knob (16 FK) at various concentrations in DMEM supplemented with 10% FBS for 45 min on ice prior to adding Ad5.eGFP, Ad5.eGFP pseudotyped with HAdV-B16 fiber, or Ad64.eGFP. The cells were incubated at 37 °C for 2 h, washed with DMEM, then incubated overnight prior to scanning all the wells for green fluorescence using a Typhoon laser scanner (Cytiva, Marlborough, MA, USA). Fiji (version 2.1.0/1.53c) was used to quantify background-subtracted total monolayer fluorescence, and the data are shown as a percent of control infection in the absence of inhibitor.

### 2.4. Viral Binding to CD46

Purified Ad64.eGFP was mixed with excess sCD46-C and 1 mM CaCl_2_, then with NiNTA-Agarose (Qiagen, Hilden, Germany) slurry. The samples were rocked for 1 h at 4 °C, then the beads were separated from the flow-through solution using gravity filtration in a Poly-Prep chromatography column (Bio-Rad) and washed three times with 4 column volumes of Dulbecco’s phosphate-buffered saline (DPBS), 10 mM imidazole. Proteins were eluted with three fractions of 1 column volume of DPBS, 0.2 M imidazole. The proteins from all the fractions were separated by SDS-PAGE using a TGX Stain-free gel (Bio-Rad) and visualized using UV activation and irradiation in a ChemiDoc MP imager (Bio-Rad).

Five nm Ni-NTA-Nanogold^®^ beads (NanoProbes, Yaphank, NY, USA) were diluted in DPBS, 1 mM CaCl_2_ mixed with sCD46-C, mixed with purified Ad64.eGFP, or mixed with sCD46-C and Ad64.eGFP. Each solution was centrifuged twice at 6000 rpm for 30 s using Ultrafree^®^-MC Centrifugal Filter Devices with a 0.1 µm pore size (MilliporeSigma, Burlington, MA, USA) to filter out material that did not bind to the virus. Volume that passed through the filter was replaced with DPBS. The treatment solutions were placed on a formvar/carbon copper grid and allowed to incubate for 10 min before replacing with 1% glutaraldehyde and 4% paraformaldehyde onto the grid to covalently crosslink the capsid components. The grids were stained with 4% sodium silicotungstate (City Chemical, West Haven, CT, USA) and air dried before imaging. The images were taken on a JEM-2100 Plus Transmission Electron Microscope (JEOL, Tokyo, Japan) at 60,000× magnification.

### 2.5. CD46 Expression on HCjE Cells

RNA was extracted from approximately 10^6^ HCjE cells using 1 mL of Trizol (ThermoFisher) reagent according to the manufacturer’s protocols. The first strand of cDNA was synthesized by mixing 4 µL SuperScript VILO Mastermix containing random primers (Invitrogen, Waltham, MA, USA) with 12 µL water, then 4 µL Trizol-extracted RNA. The mixture was incubated at 25 °C for 10 min, then 42 °C for 60 min, then 85 °C for 5 min to terminate the reaction. The cDNA was amplified with the following primers (based on [31]): PPIB_fw—TGTGGTGTTTGGCAAAGT, PPIB_rev—TGGAATGTGAGGGGAGTG, CD46_exon6_fw—TGACAGTAACAGTACTTGGGA, CD46_exon1213_rev—ATCAGTTAGGTATGTGCCTTTC, CD46_exon1214_rev—ACCATCTGCTTTCCCTTTC. The PPIB_fw and PPIB_rev pair amplified constitutively expressed PPIB. CD46_exon_6_fw forward primer binds exon 6 of CD46 for all the isoforms. The presence or omission of exon 13 dictates the tail type. CD46_exon_1213_rev amplified tail 1 isoform. CD46_exon_1214_rev amplified tail 2 isoform. Exon 8 encodes the 15 amino acid domain B of isoform BC, while omission of this exon corresponds to the C isoform. The products were amplified using Phusion High Fidelity PCR Master Mix with HF buffer (New England Biolabs, Ipswich, MA, USA) by 30 cycles of 98 °C for 10 s, 55 °C for 30 s, and 72 °C for 30 s following an initial 98 °C for 30 s step, then separated using 1.5% agarose gel electrophoresis.

Whole HCjE cell lysate was obtained by lysing a confluent 225 cm^2^ flask of cells with Mammalian Protein Extraction Reagent (M-PER; ThermoFisher Scientific). Lysate was separated on a 7.5% polyacrylamide gel and transferred to PVDF. The resulting blot was blocked with UltraCruz Blocking Reagent (Santa Cruz Biotechnology, Santa Cruz, CA, USA) at room temperature for 1 h, then incubated with a 1:1000 dilution of CD46 antibody (C-10) conjugated to Alexa Fluor 488 (Santa Cruz Biotechnology) in blocking reagent overnight at 4 °C. The blot was washed with TBS with 0.02% Tween 20 (TBS-T) and imaged using a ChemiDoc MP imager. For comparison, HeLa lysates were separated on a 7.5% polyacrylamide gel and transferred to PVDF. The membrane was blocked with 3% nonfat milk in TBS-T for 1 h, incubated with 1:1000 dilution of monoclonal rabbit anti-CD46 antibody (Abcam, Cambridge, UK), washed with TBS-T, incubated with 1:5000 anti-rabbit IgG HRP conjugate (Abcam) in TBS-T for 30 min., washed four times with TBS-T, then developed for 5 min. with SuperSignal West Pico Plus substate (ThermoFisher). The blot was imaged using a ChemiDoc MP imager. 

Confluent T75 flasks of HeLa and HCjE cells were fully detached using TrypLE Express (ThermoFisher Scientific) at 37 °C. The cells were spun down at 300× *g* for 5 min and washed once with PBS. The samples were resuspended in PBS and incubated with diluted CD46 antibody (C-10) conjugated to Alexa Fluor 488 for approximately 4 h on ice. The control cells were not exposed to the antibody. The cells were spun down and resuspended in a small amount of PBS for flow cytometry.

## 3. Results

### 3.1. CD46 Serves as a Cell Surface Receptor for HAdV-D64

Because HAdV-D64 and HAdV-D37 have identical fiber proteins and similar disease tropism, we hypothesized that HAdV-D64 would use the same receptors, CD46 and sialic acid, as HAdV-D37. It was previously shown that HAdV-D37 interaction with Chang conjunctival cells and with CD46 is dependent on the presence of divalent calcium ions [32]. Chang conjunctival cells had been used in studies [14,18,32] of HAdV-D37 interactions with its receptors but were shown to be contaminated with HeLa cervical carcinoma cells, which may have mixed its genome with or overgrown the conjunctival cells (https://www.atcc.org/products/ccl-20.2, accessed on 19 June 2024; [33]). Due to the uncertainty of the origin of Chang conjunctival cells, we chose to directly test HeLa cells, which we used to represent cells in the genitourinary tract targeted by HAdV-D64 and D37. To determine if Ad64 uses CD46 and/or sialic acid as receptors for entry into HeLa cells, we treated the cells with either 10 mM ethylene glycol-bis(β-aminoethyl ether)-N,N,N′,N′-tetraacetic acid (EGTA), which preferentially chelates and sequesters calcium(II) ions, or 10 mU neuraminidase from *Vibrio cholerae*, which enzymatically removes cell surface sialic acid sugars. We infected treated cells with an E3-deleted adenovirus type 64 containing the enhanced green fluorescent protein (eGFP) gene driven by a cytomegalovirus promoter (Ad64.eGFP) and then assessed eGFP gene delivery using flow cytometry. Neuraminidase and EGTA treatment both significantly decreased Ad64.eGFP entry (Figure 1A), indicating that HAdV-D64 entry, like HAdV-D37 entry, is dependent on the presence of sialic acid and/or calcium. Cleaving sialic acid off the HeLa cell surfaces decreased Ad64 entry by ~40%, whereas chelating calcium decreased viral gene delivery by ~97%, suggesting that viral entry is only partially dependent on sialic acid, but almost entirely dependent on the presence of calcium. 

Because calcium can have a variety of effects on adenovirus entry, we next tested to see if HAdV-D64 specifically uses CD46 to enter cells. To test this hypothesis, we incubated A549 lung carcinoma cells with increasing concentrations of recombinant fiber knob from adenovirus type 16 (16 FK), a species B adenovirus previously shown to use CD46 as a receptor [22]. As expected, the 16 FK had no effect on the entry of HAdV-C5, which has been shown to use the coxsackievirus-adenovirus receptor (CAR), and efficiently blocked HAdV-C5 pseudotyped with HAdV-B16 fiber (Ad5.F16) in a concentration-dependent manner with a 50% inhibition concentration (IC_50_) of 9.8 nM (95% confidence interval [CI], 7.3 to 13.3 nM) (Figure 1B). Sixteen FK also blocked Ad64 entry (IC_50_, 97 nM; 95% CI, 58.8 to 160.4 nM), suggesting that Ad16 and Ad64 use the same receptor, CD46, on the cell surface. Moreover, the ~10-fold higher IC_50_ of 16 FK for HAdV-64 than HAdV-5.F16 suggests a lower avidity of HAdV-64 than HAdV-16 for their common receptor. 

To substantiate this interpretation, we infected HeLa cells in the presence of calcium and a high concentration of an anti-CD46 antibody known to block HAdV-D37 entry [14]. The addition of calcium to Tris-buffered saline (TBS) substantially increased Ad64.eGFP-mediated gene delivery, but that increase can be completely abrogated by CD46 antibody (Figure 1C), suggesting that blocking CD46 inhibits calcium-dependent HAdV-D64 entry. To test whether expression of CD46 on receptor-negative cells confers susceptibility to HAdV-D64 entry, we infected Chinese hamster ovary (CHO) cells containing an expression plasmid for the BC1 or C1 isoform of human CD46 or the same plasmid containing the CD46 open reading frame inserted in the reverse orientation (RCHO) with Ad64.eGFP. Expression of CD46-BC1 or -C1 increased Ad64.eGFP infection 18-fold or 10-fold, respectively (Figure 1D). These experiments show that HAdV-D64 entry can be blocked by proteins that specifically bind CD46 (HAdV-B16 fiber knob and CD46 antibody) and that HAdV-D64 enters cells expressing CD46.

### 3.2. HAdV-D64 Binds CD46

By definition, a receptor must directly interact with HAdV-D64. To determine if CD46 meets this requirement, we investigated interactions between a soluble, C-terminally 6-His-tagged ectodomain of the C isoform of CD46 (sCD46-C) and HAdV-D64. The 6-His-tag binds specifically and tightly to divalent nickel ions. We first used sCD46-C to precipitate Ad64.eGFP particles from the solution (Figure 2A). Due to extensive N-linked and O-linked glycosylation, sCD46-C appears as a smear of ~40–50 kiloDaltons (kDa) molecular mass under SDS-PAGE (Figure 2A, left bottom panel and main panel lane 1). The HAdV-D64 hexon, the most abundant and massive protein in the capsid, appears as a band at ~110 kDa (Figure 2A, left top panel). Due to the low concentration of virus during co-precipitation, other viral proteins were not abundant enough to visualize in the gel (Figure 2A main panel lane 1). Both the 6His-tagged sCD46-C and HAdV-D64 particles were removed from the solution by nickel-nitrilotriacetic acid (NiNTA)-Agarose (lane 2), could not be washed off the beads (lanes 3–5), and eluted together by high imidazole concentration (lanes 6–8), indicating that sCD46-C binds directly to the HAdV-D64 capsid. To ensure that the apparent co-precipitation of HAdV-D64 particles was not due to nonspecific adhesion to the agarose beads [34], we mixed sCD46-C with Ad64.eGFP and 5 nm Ni-NTA NanoGold beads and then determined if the beads would be retained by a 0.1 µm centrifugal filter using transmission electron microscopy (Figure 2B). NanoGold beads easily pass through the filter if they are not bound by to a large particle like adenovirus. NanoGold bead counts per view field were much higher in the presence of virus and CD46 compared with beads alone, beads with CD46, or beads with virus (Figure 2B). Together, these results indicate that the 6-His-tagged sCD46-C can serve as a link between NiNTA and HAdV-D64 and, thus, that HAdV-D64 and CD46 form a stable complex. 

### 3.3. HAdV-D64 Uses Cell Surface CD46 as a Receptor on Human Conjunctival Epithelial Cells

Previous studies of species D adenoviruses with ocular tropism employed A549 lung epithelial cells [16,17] and human corneal cells [16,20]. To study adenoviral interactions with conjunctival cells, we used cdk4- and p53-mutated human conjunctival epithelial cells expressing human telomerase (HCjE). Expression of CD46 isoforms by immortalized HCjE cells was confirmed by RT-PCR for exon 6 and exons 12/13 (cytoplasmic tail 1) or exons 12/14 (cytoplasmic tail 2) (Figure 3A). The 45 base pair exons 7, 8, and 9 encode STP-A, -B and -C, 15 amino acid regions rich in serine, threonine, and proline. Exon 7 (STP-A) is spliced out in most tissues; the presence or absence of exon 8 results in the expression of BC or C, respectively, isoforms. BC isoform transcripts have predicted an RT-PCR amplicon of 289 (tail 1) or 286 (tail 2) base pairs, while C transcripts have predicted amplicons of 244 (tail 1) or 241 (tail 2) base pairs. RT-PCR of HCjE cells produced bands of sizes corresponding with BC1 and C1 (lane 5) and BC2 and C2 (lane 6) isoforms (Figure 3B), suggesting that alternative splicing produces all four main isoforms of CD46. Bands appeared similar in intensity between isoforms containing cytoplasmic tails 1 and 2, but the bands for BC isoforms (286/289 bp) appeared more intense than the bands for C isoforms (241/244 bp), suggesting higher expression of BC spliceosoforms. Cell surface expression of CD46 was assessed by flow cytometry using an Alexa Fluor 488-conjugated anti-CD46 antibody. Both HeLa and HCjE cells showed strong surface expression of CD46 (Figure 3C,D). Western blotting of HCjE cell lysate confirmed expression of CD46 with apparent molecular mass between 45 and 70 kDa, similar to CD46 blotting from HeLa lysates (Figure 3E). CD46 is extensively N-linked and O-linked glycosylated, producing bands of ~50–60 kDa for C isoforms and 60–70 kDa for BC isoforms that often smear into one long streak.

Having confirmed the expression of all major CD46 isoforms and the presence of CD46 on the surface of HCjE cells, we then assessed the ability of HAdV-D64 to enter HCjE cells through its known receptors, CD46 and sialic acid. Upon infection with Ad64.eGFP in the absence of inhibitors, 2.4% of the cells were GFP-positive. Treatment with sCD46-C or neuraminidase decreased Ad64.eGFP-mediated gene delivery 3.4-fold or 2.7-fold, respectively, indicating that Ad64 can use either CD46 or sialic acid as receptors on HCjE cells (Figure 4A). We expected to see a larger decrease in viral entry when both receptors were blocked compared with blocking one receptor, but we observed no statistical difference between treatments with soluble CD46 alone, with neuraminidase alone, or with both, suggesting that both sialic acid and CD46 can serve as receptors for HAdV-D64 entry into HCjE cells. We also observed that, like with HeLa cells, EGTA inhibited Ad64.eGFP gene delivery into HCjE cells by >90% (Figure 4B), indicating a strong requirement for the presence of divalent calcium for viral entry.

## 4. Discussion

Based on sequence similarity between the fiber proteins of adenoviruses associated with epidemic keratoconjunctivitis, we hypothesized that HAdV-D64 would use the same receptors, sialic acid and CD46, to adhere to the cell surface as HAdV-D37. Consistent with another recent study using A549 cells [35], we found that Ad64 entry into HeLa and HCjE cells is inhibited, at least partially, by cell surface treatment with neuraminidase (Figure 1A and Figure 3E), confirming that sialic acid plays a role as a receptor. HAdV-D64 entry can be competitively inhibited by an excess of HAdV-B16 (species B) fiber knob (Figure 1B), anti-CD46 antibody (Figure 1C), or soluble CD46 protein (Figure 3E), and can be conferred by CD46 expression on the surface of non-susceptible CHO cells (Figure 1D), clearly showing that CD46 serves as a cell surface receptor. Previous studies have shown that purified soluble HAdV-D37 fiber knob, which has an identical sequence to the fiber knob of HAdV-D64, binds directly to CD46 and that an HAdV-C5 vector pseudotyped with the HAdV-D37 fiber enters cells through CD46 [14,32], marking the fiber as the strongest candidate protein on the surfaces of HAdVs D37 and D64 to bind CD46. In our hands, binding to CD46 accounted for at least half of HAdV-D64 entry into A549 cells (Figure 1B), HeLa cells (Figure 1D), and HCjE cells (Figure 3E), making CD46 an important receptor on multiple cell lines. 

While it is not surprising that HAdV-D64 uses the same receptors as D37, our studies establish that CD46 functions as an attachment receptor on target cells associated with viral tropism, conjunctival epithelial cells, and cervical epithelial cells. Previous molecular studies on HAdV-D37 employed Chang conjunctival cells [14,18,32,36], which were later shown to be contaminated with HeLa cells during establishment (ATCC), or A549 lung epithelial cells [16,17]. HAdVs D19, D37, and D64 are adenoviruses associated not only with epidemic keratoconjunctivitis, but also with genitourinary system infections. Here, we show that HAdV-D64 can bind sialic acid and/or CD46 on immortalized human conjunctival epithelial cells and HeLa cervical carcinoma cells, two cell lines representative of targets of HAdVs D19, D37, and D64 infection. We also show that, like HeLa cervical carcinoma cells [23,37], conjunctival epithelial cells express four major spliceoforms of CD46 mRNA (Figure 3B). HCjE and HeLa cells may serve as valuable model cell lines for research on adenoviral conjunctivitis and genitourinary infections. As a protector from self-harm by the complement system [38], CD46 is widely expressed on the surfaces of almost all human cell types and tissues (The Human Protein Atlas, https://www.proteinatlas.org/ENSG00000117335-CD46, accessed on 9 September 2024; [39]). Thus, CD46 expression on the cell surface does not alone explain HAdV-D64 tropism for the conjunctiva and the genitourinary tract. 

The wide expression of CD46 and sialic acid could allow HAdV-D64 access to many cell types and tissues, making it an attractive vehicle for gene delivery. A vaccine vector based on HAdV-D64 showed higher transduction efficiency to a variety of human leukocytes and to dendritic cells [40], which matches CD46 expression on leukocytes and dendritic cells (The Human Protein Atlas, accessed on 9 September 2024; [39]). In vitro experiments show that HAdV-D64 can also efficiently enter blood endothelial cells [30] and lung [35], cervical, and conjunctival epithelial cells (this study). Determination of the CD46-binding site on the surface of the HAdV-D64 capsid will be important for the future development and alteration of the virus as a gene delivery vehicle. Previous experiments suggest that highly related HAdV-D37 binds CD46 through its fiber knob [14,32]. While adenoviruses generally use the fiber protein to bind cell surface receptors for adhesion and the penton base protein to induce internalization [41], some studies have found that the hexon protein can mediate adenoviral adhesion via molecules such as lactoferrin or Coagulation Factor IX or X [42,43,44]. Additionally, adenovirus type 56, a fellow species D adenovirus, was shown to use CD46 as an adhesion receptor, and the HAdV-D56 hexon protein, but not fiber knob, bound to CD46 protein [26]. Soluble CD46 blocked entry into A549 lung epithelial cells by a wide variety of species D adenoviruses (D13, D17, D23, D24, D25, D26, D28, D32, D38, D39, D42, D43, D45, D46, D48, and D56), but notably did not block EKC-associated HAdV-D37 [26], which has an identical fiber protein as HAdV-D64. In contrast, a recent study showing that HAdV-D37 entry into A549 cells was inhibited by knocking out CD46 [45] and our results showing that the HAdV-B16 fiber knob, known to bind CD46, blocks HAdV-D64 entry into A549 cells (Figure 1B) support the notion that CD46 can function as a receptor for HAdVs D37 and D64 on A549 cells. Alpha-defensin peptides can also mediate HAdV-D64 entry, but it is not yet clear to which viral capsid protein(s) defensins bind to increase viral entry [35]. 

The interaction between HAdV-D64 and CD46 appears to be dependent on the presence of calcium ions. Previous studies have shown that the interaction between HAdV-D37 virion and CD46 could be interrupted by the divalent cation chelator ethylene diamine tetraacetic acid, EDTA [14,32], and that calcium, but not magnesium, encouraged HAdV-D37 virion binding to cells at 1 mM concentration [32]. The calcium-specific chelator, EGTA, almost completely inhibits the entry of HAdV-D64 into HeLa and HCjE cells (Figure 1A and Figure 4B). The addition of calcium to 1 mM to Tris-buffered saline significantly increases HAdV-D64 entry into HeLa cells, and that increase can be abrogated by a CD46 antibody (Figure 1C). Calcium may also play a role in the interaction between the adenovirus penton base and cell surface integrins [46,47]. Our results suggest that the impact of calcium on Ad64 entry is partially imparted by augmenting the interaction between the HAdV-D64 virion and CD46. Although it remains unclear whether calcium influences Ad64 capsid structure, integrin function, and/or CD46 structure, calcium is clearly required for HAdVs D37 and D64 entry. This requirement suggests that the removal of calcium ions from infected or potential infected tissues can be an effective treatment or prophylaxis against adenovirus-associated ocular disease. Topical application of EDTA has already been shown as a safe and effective treatment for band keratopathy [48] and could be extended to treat or decrease transmission of epidemic keratoconjunctivitis.

## Figures and Tables

**Figure 1 viruses-16-01827-f001:**
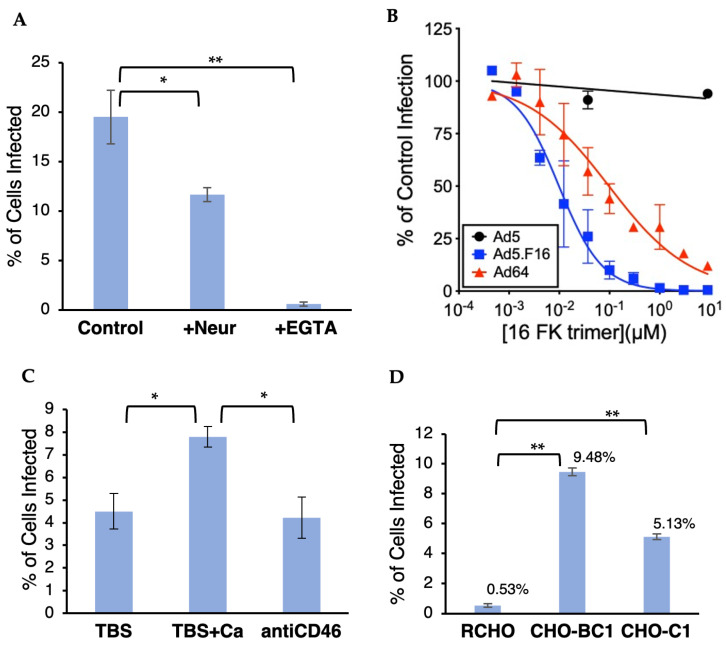
Ad64.eGFP delivers GFP gene via CD46. (**A**) Pre-treatment of HeLa cells with 10 mU neuraminidase from *Vibrio cholerae* to remove cell surface sialic acid or 10 mM EGTA to chelate calcium ions prior to incubation with Ad64.eGFP significantly decreased susceptibility to viral entry as determined by cell fluorescence compared to uninfected cells (Analysis of variance followed by Tukey’s test (ANOVA-T), n = 3 for each treatment). Error bars represent standard deviations. (**B**) HAdV-B16 fiber knob (16 FK) blocked Ad5 pseudotyped with Ad16 fiber (Ad5.F16) and HAdV-D64 (Ad64), but not HAdV-C5 (Ad5) gene delivery into HeLa cells in a concentration-dependent manner. Experiments were performed in duplicate. (**C**) Anti-CD46 antibody blocked calcium-dependent entry into HeLa cells (ANOVA-T, n = 3 for each treatment). (**D**) Expression of CD46 BC1 or C1 isoform on CHO cells conferred susceptibility to Ad64.eGFP entry (ANOVA-T, n = 3 for each treatment). Single asterisk, *p* < 0.05; double asterisk, *p* < 0.0001.

**Figure 2 viruses-16-01827-f002:**
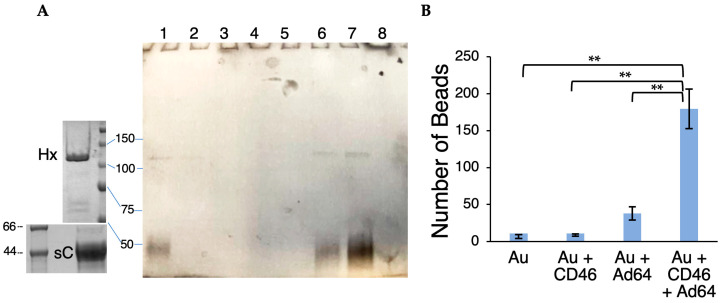
Direct binding of HAdV-D64 to soluble CD46. (**A**) Ad64.eGFP particles were co-precipitated with His-tagged soluble CD46 (sC) using Ni-NTA Agarose then subject to SDS-PAGE. Main panel lanes: 1, CD46 (~45–50 kDa) mixed with Ad64.eGFP; 2, flowthrough from NiNTA Agarose, 3–5 wash fractions, 6–8 elution fractions. Hx, HAdV-D64 hexon. Gel images of ultracentrifugation-purified Ad64.eGFP (top left) and purified sCD46-C (bottom left) are shown to the left for comparison. Masses of molecular standards are shown in kDa. (**B**) 5 nm Ni-NTA-NanoGold beads (Au) were incubated with CD46, Ad64.eGFP particles, or both, filtered through a 0.1 µm filter, visualized using transmission electron microscopy at 60,000× magnification, and counted. Only the presence of the large adenovirus particle and CD46 led to substantial retention of NanoGold beads in the filter (ANOVA-T, n = 3 for each treatment). Double asterisks, *p* < 0.0001.

**Figure 3 viruses-16-01827-f003:**
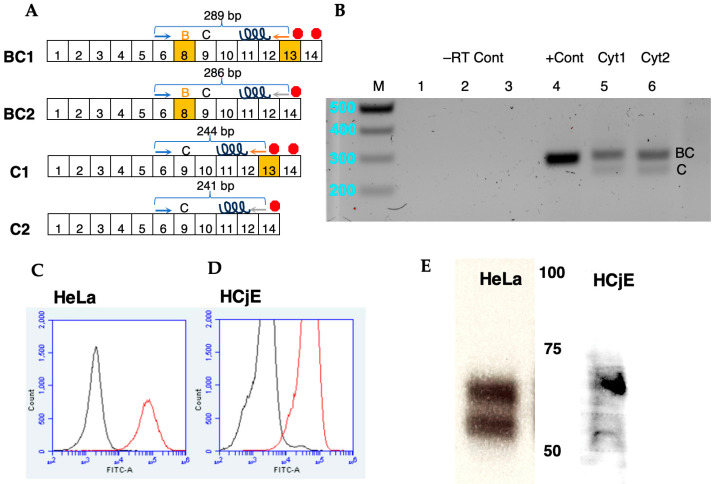
Expression of CD46 isoforms on human conjunctival epithelial cells. (**A**) Diagram of four main CD46 spliceoforms. CD46’s exons are shown as numbered boxes (not to scale). Splicing out exon 8, encoding the STP-B domain, produces C isoforms (C1 and C2), and splicing out exon 13 produces Cyt2 isoforms (BC2 and C2) containing an alternate cytoplasmic tail with a stop codon (red octogons). The locations of forward primer CD46_exon6_fw (blue arrow) and reverse primers CD46_exon1213_rev (orange arrow) and CD46_exon1214_rev (gray arrow) are shown above the exons to which they match. The transmembrane helix is encoded by exons 11 and 12 and shown as a spiral. (**B**) Agarose gel electrophoresis of RT-PCR products from HCjE cells using gene and isoform specific primers. Lanes: M, Marker; 1 & 4, constitutively expressed PPIB gene; 2 & 5, CD46 isoforms containing cytoplasmic tail 1 (Cyt1); 3 & 6, CD46 isoforms containing tail 2 (Cyt2). No RT control (–RT Cont) samples for lanes 1–3 contained no reverse transcriptase. Representative flow cytometry plots of (**C**) HeLa or (**D**) HCjE cells stained with AlexaFluor488-labeled CD46 antibody (red lines). Black lines are unlabeled cells. (**E**) Western blotting of HeLa and HCjE cell lysate using Alexa Fluor 488-labeled CD46 antibody shows CD46 protein expression at ~50–70 kDa (molecular weight markers, center).

**Figure 4 viruses-16-01827-f004:**
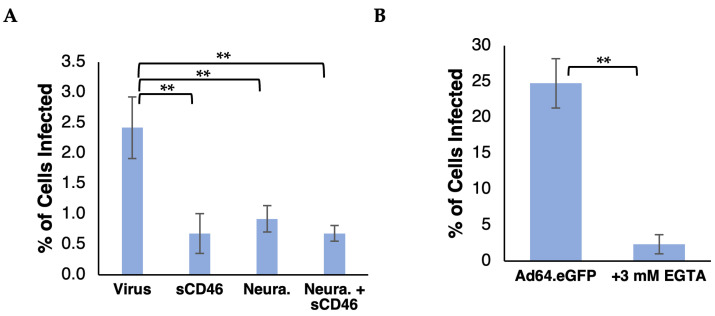
HAdV-D64 can enter human conjunctival epithelial cells. (**A**) Pre-treatment of HCjE cells with soluble CD46-C (sCD46) and/or neuraminidase inhibited Ad64.eGFP viral entry, as determined by expression of green fluorescence of eGFP transgene (ANOVA-T, n = 5). (**B**) Ad64.eGFP vector entry into HCjE cells in Keratinocyte-SFM in the presence or absence of EGTA, a calcium chelator (ANOVA-T, n = 3). ** denotes *p* < 0.001.

## Data Availability

The data underlying this article are available in this article.

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
