# Peer review of "CD46 Is a Protein Receptor for Human Adenovirus Type 64"

_viruses, 2024, doi:10.3390/v16121827_

Round 1
Reviewer 1 Report
Comments and Suggestions for Authors
Dear authors,
I would like to commend your effort in conducting and presenting your research. However, after thoroughly reviewing your manuscript, I believe it is not ready for publication in Viruses. I understand that receiving this feedback is not easy, but I hope my comments will help you improve your study for future submissions.
First, the overall rationale behind your study is unclear. You explore whether sialic acids and CD46 are receptors for HAdV-D64, despite the fact that HAdV-D37 and HAdV-D64 have identical fibers. In your manuscript, you mention twice that they share large homology; this is not accurate, as their fibers are 100% identical. Given this, the investigation of these receptors for HAdV-D64 appears redundant, as the fiber knob is most likely the key determinant for receptor interactions. This study may be more appropriate as part of a larger investigation or a conference presentation but does not, in its current form, provide sufficient novelty or depth for a peer-reviewed journal article.
Additionally, the use of only one conjunctival and one cervical cell line is insufficient to make broad conclusions about HAdV-D64's receptor usage. This limitation raises concerns about the study's scope and the generalizability of the findings. Furthermore, the infection efficiency in HeLa cells is notably low, with only about 20% infection at a high MOI of 1000. This outcome suggests possible issues with the cell line's susceptibility to HAdV-D64, and additional cell lines would be required to validate your conclusions. The infection of HCjE cells, which resulted in only 2.5% infected cells, also raises questions, particularly since the MOI used in these experiments was not mentioned in your Materials and Methods. While CD46 contributes to HAdV-D64 entry, the low infection rate suggests that it might not be the primary receptor, and other receptors could have a more prominent role.
Additionally, you mentioned that in the study aimed at establishing the receptors for HAdV-D37, Chang conjunctival cells were used, which were later shown to be contaminated with HeLa cells. However, it is unclear what conclusion readers should draw from this. Should receptor usage of Ad37 be reassessed using a new cell line, or can we still trust the existing data? Alternatively, should HeLa cells be tested separately? Or are you suggesting that HeLa cells are also susceptible to Ad37 infection, just like conjunctival cells? Please clarify this point in your revised manuscript for future submissions.
Moreover, I recommend that all three figures in the manuscript be revised:
- The brackets indicating significant differences between columns should not intersect with the columns or error bars themselves.
- Fonts within each figure should be consistent.
- Axis labels should be aligned and clearly legible.
- The use of color should be minimized to avoid redundancy.
- Markers should be properly displayed; for instance, Figure 2 is missing a marker.
I hope this feedback will guide you in strengthening your manuscript for future submission.
Author Response
Reviewer 1
Comment 1: “First, the overall rationale behind your study is unclear. You explore whether sialic acids and CD46 are receptors for HAdV-D64, despite the fact that HAdV-D37 and HAdV-D64 have identical fibers. In your manuscript, you mention twice that they share large homology; this is not accurate, as their fibers are 100% identical. Given this, the investigation of these receptors for HAdV-D64 appears redundant, as the fiber knob is most likely the key determinant for receptor interactions.”
Response 1: We appreciate Reviewer 1’s skepticism about our study’s rationale. It is true that HAdV-D37 and HAdV-D64 have identical fiber proteins, and we have changed the wording in our manuscript in the abstract (page 1, line 15) and introduction (page 2, line 71) to state that the fiber proteins are identical. We have also added several sentences in the introduction (page 2, lines 61-70) to make the rationale for our study clearer. Traditionally adenovirologists have assigned the fiber knob as “the key determinant for receptor interactions”, but a recent report showed that species D adenoviruses can interact with their receptors using their hexons. Because HAdV-D64 differs from D37 substantially in its hexon genes, it was not known what the receptor(s) is for D64 nor what capsid protein is important for binding to the receptor(s). The alterations to the abstract and introduction help make the case that the fiber may not be the only determinant for receptor usage and that our study is not redundant as a result.
Comment 2: “the use of only one conjunctival and one cervical cell line is insufficient to make broad conclusions about HAdV-D64's receptor usage. This limitation raises concerns about the study's scope and the generalizability of the findings…. While CD46 contributes to HAdV-D64 entry, the low infection rate suggests that it might not be the primary receptor, and other receptors could have a more prominent role.”
Response 2: We note that our study also showed that HAdV-D64 uses CD46 as a receptor on A549 lung epithelial cells (Figure 1B) in addition to the conjunctival and cervical cell lines noted by Reviewer 1. Few immortalized cell lines representing a key tissue for epidemic keratoconjunctivitis exist: Chang conjunctival cells, which as we noted is contaminated with HeLa cells, IOBA-NHC cells, which have shown signs of senescence, and HCjE cells, which were studied here in our experiments; (Garcia-Posadas et al., 2022 Characterization and functional performance of a commercial human conjunctival epithelial cell line. Experimental Eye Research. 223:109220). Garcia-Posadas et al. describe the recent development of a commercial conjunctival cell line in 2022. While we would like to study HAdV-D64 on more cell lines in the future, we believe that showing HAdV-D64 entry into HCjE, HeLa, and A549 cells via CD46 constitutes a novel and substantial contribution to the literature using the important available model cells. We accept that the availability of cell lines representing conjunctiva is a limitation.
We acknowledge that infection rates of HeLa and HCjE cells described in our manuscript are low. We note that particle-to-PFU ratios for well-studied animal viruses range from 10-10,000 and for adenoviruses 20-100 (Principles of Virology, Volume 1). Concentrations of virus were titrated to produce positive fluorescence readings between 2-25% to limit saturating cells and minimize superinfection. We agree that HAdV-D64 may use other receptors. In fact, our results indicate that sialic acid also plays an important role as a receptor.
Comment 3: “Additionally, you mentioned that in the study aimed at establishing the receptors for HAdV-D37, Chang conjunctival cells were used, which were later shown to be contaminated with HeLa cells. However, it is unclear what conclusion readers should draw from this. Should receptor usage of Ad37 be reassessed using a new cell line, or can we still trust the existing data? Alternatively, should HeLa cells be tested separately? Or are you suggesting that HeLa cells are also susceptible to Ad37 infection, just like conjunctival cells? Please clarify this point in your revised manuscript for future submissions.”
Response 3: Our manuscript attempts to make the case that there are two cell lines, HCjE and HeLa cells, that make better models than Chang conjunctival cells for infections by this group of species D adenoviruses. The main reason is that the identity of Chang cells is unclear due to contamination with HeLa cells. It was previously shown that HeLa cells are susceptible to entry by an adenovirus vector containing the HAdV-D37 fiber and that a CD46 antibody could block entry by that same vector (Wu et al., 2004, JVI 78:3897-3905). Because our manuscript focuses on HAdV-D64, we have not revisited the experiments on HAdV-D37, but all evidence points to the possibility that Chang conjunctival cells were overgrown by HeLa cells and functionally are the same as HeLa cells. We have added several lines (page 4, lines 224-227) to clarify why we chose to study HAdV-D64 using HeLa cells instead of Chang cells as a model cell line.
Comment 4: “Moreover, I recommend that all three figures in the manuscript be revised:
- The brackets indicating significant differences between columns should not intersect with the columns or error bars themselves.
- Fonts within each figure should be consistent.
- Axis labels should be aligned and clearly legible.
- The use of color should be minimized to avoid redundancy.
- Markers should be properly displayed; for instance, Figure 2 is missing a marker.”
Response 4: We thank the reviewer for noting the formatting issues associated with our figures. The brackets indicating significant differences have been repositioned. Fonts and address labels have been made consistent between our bar graphs. We have added molecular mass markers to our gel pictures.
Reviewer 2 Report
Comments and Suggestions for Authors
Manus by Wu deals with CD46 as the receptor for HAdV64. As such, the manuscript is well written and provides some extra data regarding the CD46-HAdV64 interaction in different cell models, although its format is rather a short report than a research article. Actually, one should give credit for a very detailed M&M section, which assures me that the experiments have been well-planned and can be repeated.
Anyhow, I do have some comments that the authors should address:
1) In general the Figures should be rechecked as some of the panels are not aesthetically pleasing (e.g., Fig. 1D (overlapping significance lines), Fig. 2A (lack of Mw), Fig. 3A (not complete image, difficult understand).
2) p38 and thereon: it has to be type not genotype, since the classification is based on both serotyping and genotyping.
3) P229: and Figure 1D:Expression of CD46-BC1 or -C1 increased Ad64.eGFP infection 18-fold or 10-fold, respectively. The figure is of bad quality due to overlapping significance line, fix it. Also, it would be good to add the exact % of infected cells to the graph as it will make it easier to understand the authors claim for 18- and 10-fold effects.
4) P257 and Fig. 2A: Since the SDS-Page lacks Mw, it is very difficult to tell the origin of the proteins. Furthermore, since it is not a western blot, and lacks lanes with only virus or sC, it makes it kind of guesswork about the bands origin. At least including the Mw would make it more scientific and trustworthy.
5) P288-P309, Fig. 3A: The authors describe the CD46 splicing in details (maybe even in a bit confusing way), but do not add the corresponding info to the Fig. 3A. Why not to indicate BC and C isoforms with the arrows on the blot? Also, it is confusing using the word "tail". If there are isoforms, then the respective isoform should have defined tail, or I am missing something? If the authors want (although I feel it is out of the scope of the manus), they could provide the drawing of the CD46 alt. splicing pattern.
6) P304-P305, Fig. 3D: It is way to strong statement that "HeLa and HCjE cells showed strong surface expression of CD46." Since the images are from different gels and there are no loading controls, this statement can not be made.
7) P367: is not that an overstatement that the authors detect all 4 CD46 isoforms? The Fig. 3A does not show it.
8) P401. The statement "Our results suggest that the impact of calcium on Ad64 entry is partially imparted by augmenting the interaction between the HAdV-D64 virion and CD46" is only partially correct as the authors did the experiment only in HeLa (Fig. 1A), and not in HCjE cells. Was there a particular reason why it was not done in HCjE? Including that data would make the statement stronger.
Author Response
Reviewer 2
Comment 1: “1) In general the Figures should be rechecked as some of the panels are not aesthetically pleasing (e.g., Fig. 1D (overlapping significance lines), Fig. 2A (lack of Mw), Fig. 3A (not complete image, difficult understand).”
Response 1: We thank the reviewer for noting the formatting issues associated with our figures. The brackets indicating significant differences have been repositioned. Fonts and address labels have been made consistent between our bar graphs. We have added molecular mass markers to our gel pictures.
Comment 2: “2) p38 and thereon: it has to be type not genotype, since the classification is based on both serotyping and genotyping.”
Response 2: The reviewer is correct in the use of the word “type” to describe adenovirus classification. We have replaced all references to “genotype” or “serotype” with “type”.
Comment 3: “3) P229: and Figure 1D:Expression of CD46-BC1 or -C1 increased Ad64.eGFP infection 18-fold or 10-fold, respectively. The figure is of bad quality due to overlapping significance line, fix it. Also, it would be good to add the exact % of infected cells to the graph as it will make it easier to understand the authors claim for 18- and 10-fold effects.”
Response 3: We have fixed the overlapping significance lines and added the % of infected cells to the graph as requested.
Comment 4: “4) P257 and Fig. 2A: Since the SDS-Page lacks Mw, it is very difficult to tell the origin of the proteins. Furthermore, since it is not a western blot, and lacks lanes with only virus or sC, it makes it kind of guesswork about the bands origin. At least including the Mw would make it more scientific and trustworthy.”
Response 4: We have added the molecular mass markers to Figure 2 and supplemented the figure with SDS-gel images of purified HAdV-D64 and soluble CD46-C next to the protein bands for comparison.
Comment 5: “5) P288-P309, Fig. 3A: The authors describe the CD46 splicing in details (maybe even in a bit confusing way), but do not add the corresponding info to the Fig. 3A. Why not to indicate BC and C isoforms with the arrows on the blot? Also, it is confusing using the word "tail". If there are isoforms, then the respective isoform should have defined tail, or I am missing something? If the authors want (although I feel it is out of the scope of the manus), they could provide the drawing of the CD46 alt. splicing pattern.”
Response 5: We thank the reviewer for noting the lack of clarity in the RT-PCR data (formerly Figure 3A). As requested, we have indicated BC and C spliceoform products in the figure (now Figure 3B). The usage of the word “tail” follows the all other published papers on CD46. To help make sense of the RT-PCR data, we have created a diagram of the 4 main spliceoforms of CD46 to illustrate the locations of the primers as well as which exons the primers anneal to (new Figure 3A). We hope the diagram will help the reader better understand the experimental results.
Comment 6: “6) P304-P305, Fig. 3D: It is way to strong statement that "HeLa and HCjE cells showed strong surface expression of CD46." Since the images are from different gels and there are no loading controls, this statement can not be made.”
Response 6: In the original manuscript draft the reference to strong surface expression of CD46 was meant to direct the reader to the flow cytometry data that directly assess surface protein expression. We have moved the reference to Figures 3C and 3D (formerly Figures 3B and 3C) right behind the phrase “strong surface expression of CD46” on page 8 line 346. We apologize for the confusion.
Comment 7: “7) P367: is not that an overstatement that the authors detect all 4 CD46 isoforms? The Fig. 3A does not show it.”
Response 7: Figure 3B (formerly Figure 3A) shows our detection of 4 distinct mRNAs associated with CD46. We have changed page 10, lines 519-521, to emphasize that the data is specific to mRNA spliceoforms. Figure 3B (formerly Figure 3A) is now supported with a new Figure 3A that illustrates how the four spliceoforms can be detected using forward and reverse primers described in the Materials and Methods section.
Comment 8: “8) P401. The statement "Our results suggest that the impact of calcium on Ad64 entry is partially imparted by augmenting the interaction between the HAdV-D64 virion and CD46" is only partially correct as the authors did the experiment only in HeLa (Fig. 1A), and not in HCjE cells. Was there a particular reason why it was not done in HCjE? Including that data would make the statement stronger.”
Response 8: We thank the reviewer for pointing out the omission of an experiment addressing the role of calcium in the interaction between HAdV-D64 and HCjE conjunctival epithelial cells. Upon receiving the review, we thawed and grew HCjE cells and tested whether EGTA, a calcium-specific chelator, could inhibit HAdV-D64 entry. Those data are presented in a new Figure 4B, and we moved the former Figure 3E to a new Figure 4A, thereby grouping the two bar graphs on HAdV-D64-mediated gene delivery together into a new figure 4. We note that the HCjE infections were performed in Keratinocyte Serum Free Medium (KSFM), which already contains ~0.12 mM calcium ion. The 3 mM EGTA we used should be sufficient to sequester the vast majority of the calcium from the KSFM. We appreciate the reviewer for suggesting the experiment and making our manuscript stronger.
Reviewer 3 Report
Comments and Suggestions for Authors
Wu et al. present an interesting study that confirms the role of CD46 as receptor for adenovirus type 64, that has been hypothesized based on the recombinant origin of the fiber protein from a recombinant parent type 37, which has been previously confirmed to use CD46 as a receptor. The study went beyond this confirmation and explored deeper possible implications and the role of calcium in the internalization of the virus particles. I think this report will be an excellent reference in the characterization of adenovirus receptors and developing treatments to stop adenovirus infections in ocular tissues.
My only minor concern, which is easy to address, is that the authors omitted from the abstract the recombinant connection between Adenovirus type 64 and type 37; similarly in the introduction they get to the point to recognize high sequence homology, but it is until the results that they state that the fibers from both types are identical. I think they can recognize those facts in the abstract and the introduction without losing the novelty and meaningfulness of the rest of the study.
Author Response
Reviewer 3
Comment: “My only minor concern, which is easy to address, is that the authors omitted from the abstract the recombinant connection between Adenovirus type 64 and type 37; similarly in the introduction they get to the point to recognize high sequence homology, but it is until the results that they state that the fibers from both types are identical. I think they can recognize those facts in the abstract and the introduction without losing the novelty and meaningfulness of the rest of the study.”
Response to Reviewer 3: We thank the reviewer for noting our omission of the genetic similarities between HAdV-D37 and D64. As requested, we have added specific language to the abstract (page 1, lines 14-16) to note the genetic relationship between the two types and noted in several places in the manuscript that the two types have identical fiber proteins (page 1 line 15; page 2, line 71; page 5, line 220).
Round 2
Reviewer 1 Report
Comments and Suggestions for Authors
Dear authors,
Thank you for your detailed responses to my previous comments. I appreciate the additional work you have put into addressing my points. However, I still find that the scope of determining whether CD46 is a cellular receptor is not sufficiently relevant to support the entire paper for HAdV-D64. This investigation might be more appropriate as part of a broader study, especially if you could demonstrate which specific capsid protein is responsible for CD46 binding.
In the revised manuscript, you stated, "...creating uncertainty about whether HAdV64 also uses CD46 to enter target cells and, if so, through which capsid protein." However, the specific protein involved in binding remains unidentified. As demonstrated for HAdV-D37, it is known to utilize sialic acids for cellular entry via the fiber knob rather than the hexon. Given the identical fiber structures between HAdV-D64 and HAdV-D37, it is reasonable to deduce that HAdV-D64 likely also utilizes sialic acid for cell entry. Additionally, previous studies (J Virol. 78.8: 3897-3905, 2004) have shown that the recombinant Ad37 fiber knob can bind CD46's extracellular domain, suggesting that HAdV-D64 could use CD46 similarly.
You also noted, “In contrast, our results showing that the HAdV-B16 fiber knob, known to bind CD46, blocks HAdV-D64 entry into A549 cells and that soluble CD46 blocks entry into conjunctival epithelial cells suggests that the differences between the penton base and hexon genes of HAdV-D64 and D37 may result in differential receptor usage.” However, directly comparing HAdV-D64 infection in HCjE cells with HAdV-D37 infection in A549 cells presents challenges since these cell types differ significantly, particularly regarding their surface receptor composition. This disparity limits the validity of any conclusions drawn regarding differential receptor usage. Additionally, as highlighted in my previous comments, your use of a multiplicity of infection (MOI) exceeding 1000 resulted in only 2.5% of cells being infected, casting doubt on the robustness of the findings.
In summary, while your work offers valuable insights into adenovirus receptor interactions, the current dataset does not, in my opinion, support a standalone publication.
Author Response
Comment 1: Thank you for your detailed responses to my previous comments. I appreciate the additional work you have put into addressing my points. However, I still find that the scope of determining whether CD46 is a cellular receptor is not sufficiently relevant to support the entire paper for HAdV-D64. This investigation might be more appropriate as part of a broader study, especially if you could demonstrate which specific capsid protein is responsible for CD46 binding.
Response 1: While our results showing that CD46 and sialic acid function as receptors for HAdV-D64, a mostly uncharacterized adenovirus type with an identical fiber protein as HAdV-D37, is not entirely surprising, we do believe that our manuscript merits publication in Viruses. Importantly, we show that both cell surface molecules can function as receptors on the surface of human conjunctival epithelial cells and human cervical epithelial cells, representative cell lines of target tissues for HAdV-D19, D37, and D64. The cell surface expression of CD46 protein and production of four major CD46 spliceoforms are, to our knowledge, novel results. Lastly, we present data on the efficacy of EGTA in blocking viral entry, making it a potential antiviral or preventative measure against epidemic keratoconjunctivitis.
Comment 2: In the revised manuscript, you stated, "...creating uncertainty about whether HAdV64 also uses CD46 to enter target cells and, if so, through which capsid protein." However, the specific protein involved in binding remains unidentified. As demonstrated for HAdV-D37, it is known to utilize sialic acids for cellular entry via the fiber knob rather than the hexon. Given the identical fiber structures between HAdV-D64 and HAdV-D37, it is reasonable to deduce that HAdV-D64 likely also utilizes sialic acid for cell entry. Additionally, previous studies (J Virol. 78.8: 3897-3905, 2004) have shown that the recombinant Ad37 fiber knob can bind CD46's extracellular domain, suggesting that HAdV-D64 could use CD46 similarly.
Response 2: We agree with the reviewer that HAdV-D64 has an identical fiber protein to HAdV-D37 and, thus, should also bind the same receptors that are bound by HAdV-D37’s fiber knob, sialic acid and CD46. A motivation of our study is to test if this is the case on target cells, and our manuscript describes the results that support this hypothesis. We do not currently have data on direct interactions between D64 fiber knob and CD46, so we have chosen not to emphasize this possibility. We recognize studies that have shown direct interaction between the D37 fiber knob and CD46 (lines 401-405) in support of the possibility that HAdV-D64 binds to CD46 through its fiber knob.
Comment 3: You also noted, “In contrast, our results showing that the HAdV-B16 fiber knob, known to bind CD46, blocks HAdV-D64 entry into A549 cells and that soluble CD46 blocks entry into conjunctival epithelial cells suggests that the differences between the penton base and hexon genes of HAdV-D64 and D37 may result in differential receptor usage.” However, directly comparing HAdV-D64 infection in HCjE cells with HAdV-D37 infection in A549 cells presents challenges since these cell types differ significantly, particularly regarding their surface receptor composition. This disparity limits the validity of any conclusions drawn regarding differential receptor usage.
Response 3: This sentence was intended to explain the discrepancy between a study that showed that soluble CD46 does not block HAdV-D37 entry into A549 cells (Persson et al., PNAS 2021, 118, e2020732118) and our own results. To avoid the complication of different cell lines and viruses, we have removed the mention of HCjE cells here and replaced it with a reference to a recent article showing that HAdV-D37 enters CD46-KO A549 cells poorly (Tsoukas et al., Cells 2022, 11, doi:10.3390/cells11050841). This article and our own results both show that CD46 can function as a receptor for EKC-associated species D HAdVs on A549 cells. (Lines 449-453)
Comment 4: Additionally, as highlighted in my previous comments, your use of a multiplicity of infection (MOI) exceeding 1000 resulted in only 2.5% of cells being infected, casting doubt on the robustness of the findings.
Response 4: After receiving the first round of comments from our three reviewers, we followed up our earlier experiments with an experiment testing infection of HCjE cells by Ad64.eGFP in the presence and absence of the calcium chelator EGTA. We were able to achieve a 10-fold higher level of infection (~25%) at an MOI of 2000 (Figure 4B). The reason for the discrepancy in infection rates is unknown, but possibly due to using a substantially lower density of cells in our follow-up experiments.
Reviewer 2 Report
Comments and Suggestions for Authors
The authors have replied well to my comments, and I am happy about the manuscript modifications. No further questions from my side. Well done, and good manuscript in the adenovirus field.
Author Response
N/A